# Development of a Machine-Learning Model of Short-Term Prognostic Prediction for Spinal Stenosis Surgery in Korean Patients

**DOI:** 10.3390/brainsci10110764

**Published:** 2020-10-22

**Authors:** Kyeong-Rae Kim, Hyeun Sung Kim, Jae-Eun Park, Seung-Yeon Kang, So-Young Lim, Il-Tae Jang

**Affiliations:** 1Nanoori Medical Research Institute, Nanoori Hospital Gangnam, Seoul 06048, Korea; keyelkim@gmail.com (K.-R.K.); parkje16@hanmail.net (J.-E.P.); dlathdud3110@gnnanoori.co.kr (S.-Y.L.); 2Department of Neurosurgery, Nanoori Hospital Gangnam, Seoul 06048, Korea; neurospinekim@gmail.com; 3Department of Anesthesia and Pain Medicine, Nanoori Hospital Gangnam, Seoul 06048, Korea; 54kkang@naver.com

**Keywords:** machine learning, prediction, pilot study, spinal surgery, Korean

## Abstract

Background: In this study, based on machine-learning technology, we aim to develop a predictive model of the short-term prognosis of Korean patients who received spinal stenosis surgery. Methods: Using the data obtained from 112 patients with spinal stenosis admitted at N hospital from February to November, 2019, a predictive analysis was conducted for the pain index, reoperation, and surgery time. Results: Results show that the predicted area under the curve was 0.803, 0.887, and 0.896 for the pain index, reoperation, and surgery time, respectively, thereby indicating the accuracy of the model. Conclusion: This study verified that the individual characteristics of the patient and treatment characteristics during surgery enable a prediction of the patient prognosis and validate the accuracy of the approach. Further studies should be conducted to extend the scope of this research by incorporating a larger and more accurate dataset.

## 1. Introduction

Machine learning (ML) is a field of artificial intelligence that comprises various algorithms allowing the self-learning of data patterns using a computer [1]. Therefore, the knowledge acquired by the machine attains a practical influence, while being used in the related decision-making process and predictions of future outcomes [2].

Recently, ML algorithms have been applied for the development of predictive models in the field of medicine owing to the emergence of big data, development of various algorithms, and gradual improvement in the computational power [3]. Several ML techniques, such as the gradient boosting machine, decision-making tree, random forest, and neural network, have already been proven effective for the treatment of neurosurgical patients from the viewpoint of a greater predictive power than conventional statistical modelling techniques [4].

The number of patients with spinal stenosis in Korea has increased owing to an increase in the average life expectancy. According to the statistical survey of the Health Insurance Review and Assessment Service in Korea, the number of patients with spinal stenosis has increased substantially from 1,283,861 in 2014 to 1,649,222 in 2018 [5]. For treatment of spinal stenosis, non-surgical therapies are generally recommended and have been found to be effective; however, several studies have reported that the decompression surgery is more advantageous than non-surgical therapies [6,7].

The prognosis for the surgical treatments is hard to predict due to the requirement of a systematic and objective preoperative consultation for identifying the psychological factors of the patient related to surgery [8]. Although previous studies have reported several factors that affect the outcome of spinal stenosis surgery [9,10], it is extremely challenging to predict a single, unified result that reflects the myriad individual patient characteristics. If an ML-based predictive model incorporating all such essential and complex factors could be developed, it would likely enable both the patients and surgeons to share clinically valuable results.

It is expected that this novel technology will enable surgeons to identify patients who would benefit the most from surgical treatment and predict the prognosis and any side effects more accurately than the conventional technologies. These results will also help in preoperative decision-making of the patient based on the objective data. Therefore, the proposed study aims to develop an ML model for predicting the range of short-term endpoints from the viewpoint of pain experienced by patients who have received decompression surgery for spinal stenosis. The resulting ML model is expected to allow a predictive analysis before surgery and determine its accuracy.

## 2. Materials and Methods

### 2.1. Data Summary

The subjects in this study were 112 patients who received endoscopic and open surgery at the Spine center of N Hospital located in Seoul, South Korea, from February to November, 2019. All the patients received either single or multi-level decompression surgery. For data collection, adult patients, whose standard data were recorded without omission, were included in this study In addition, there is no informed consent form because our research is a retrospective study using patient’s medical records. This retrospective study was conducted in accordance with the Declaration of Helsinki. The protocol was approved by the Ethics Committee of Nanoori Hospital Gangnam (NR-IRB 2020-021).

### 2.2. Data Collection

The clinical and radiological data of the subjects were collected on the day of the first outpatient visit. According to the heath care procedures at the hospital, the subjects received MRI and clinical tests. The 13 parameters of the collected data were gender, age, height (cm), weight (kg), BMI, smoking (active/non-active), alcohol drinking (yes/no), medical cost affordability (high/moderate/low), surgical segment, anesthesia grade based on the American Society of Anesthesiologist score, single or multi-level decompression, index level, and the result of the patient-reported outcome measure (PROM).

#### 2.2.1. Patient-Reported Outcome Measure (Numeric Rating Scale)

The PROM was set based on a Numeric Rating Scale (NRS) with a range of 0–10, and it means the degree of suffering from radiating pain. To conduct PROM, a standardized questionnaire was used to collect the data prior to surgery, immediately after surgery, one day after surgery, upon discharge, and one month after the surgery. In this study, the short-term surgery outcome was defined as achieving the threshold of the minimum clinically important difference after surgery; therefore, the short-term clinical success was defined as a 30% decrease in the NRS than that reported by the patient before surgery [11].

#### 2.2.2. Incidence of Reoperation

The incidence of reoperation in all subjects was applied, and the data required to predict the probability of reoperation were collected.

#### 2.2.3. Surgery Time

The surgery time in all subjects was traced in this study, where the criteria was set as the patients who received surgery lasting ≥200 min. This cut-off was determined based on the observed distribution in the samples collected for this study. An analysis of the surgery time is essential, because a relatively long surgery time indicates a high probability of a prolonged length of stay [12].

### 2.3. Empirical Analysis

Herein, the continuous data were expressed based on the mean and standard deviation, whereas the categorical data were expressed in eigenvalue and percentage. Data, wherein one or more parameters were omitted, were excluded from the analysis. The collected data were randomly divided into 70% for training and 30% for the evaluation of the ML model. The ML model with the training dataset was trained using a bootstrapping model, and the training result was validated. Furthermore, for each parameter, seven ML algorithms, including random forest, XGBoost, Bayesian generalized linear model, decision-making tree model, k-cluster analysis, logistic regression analysis, and neural network analysis, were tested. Based on the results of the training and evaluation of each model, the algorithm with the greatest explanatory power was determined by comparing the area under the curve (AUC) and selecting the one with the largest value.

The final model selected was tested based on the evaluation dataset for an internal validation, and all the previously described explanatory variables were used as an input to all models for analysis. For any imbalance in the model training, the synthetic minority oversampling technique was applied to the training dataset to ensure the robustness of the model. This was because the model training using data that substantially vary in the numbers of each class causes an excessive classification of patterns into multiple categories, which would thus affect the model performance [13]. In addition, the Brier score was used to evaluate the parameters that verify the accuracy of the predicted results. The Brier score ranges between zero and one, and a value closer to zero indicates more accurate results [14]. For all analyses, STATA ver. 15. MP was used.

## 3. Results

The data obtained from 112 patients were collected through the patient analysis and cleaning from February to November, 2019. All the data were included in the training model for subsequent trainings. The number of males in this dataset was 50, accounting for 44.6% of the total dataset, with a mean age of 60.4 years. The details of the demographic characteristics and the information regarding surgery are listed in Table 1.

### 3.1. Patient-Reported Outcome Measure

The rate of reduction in postoperative NRS indicated a clinically significant improvement in 80.4% of the patients. The percentage of patients who received a reoperation was 5.3%, whereas the percentage of patients with a surgery time of >200 min was 17.9% (Table 2). In addition, the individual accuracy of the developed predictive models was 83%, with an AUC of 0.860.

These parameters imply that each model differs from the viewpoint of the predicted values and accuracy depending on the parameters, and an over-fitting does not occur based on the comparison of the training and evaluation results [15]. The Brier score is the mean of the square of the difference between the observed and predicted results obtained through an equation comparing the overall accuracy (Table 3).

### 3.2. Reoperation

The incidence of reoperation occurred in six patients (5.3%), predicted with 93% accuracy with an AUC of 0.952. No over-fitting was observed for the reoperation predictive model.

### 3.3. Surgery Time

Surgery time of ≥200 min was recorded for 20 patients (17.9%). The prediction of prolonged surgery time showed an accuracy of 95%, but the predicted AUC was 0.975. For this model based on the random forest algorithm, the sensitivity (96%) and positive predicted value (98%) were both high, and the sensitivity was 90%.

A graph representing the AUC of each model can be found below (Figure 1).

## 4. Discussion

In this study, ML-based prognosis predictive models were developed using the data obtained from 112 patients who received spinal stenosis surgery, and the prediction accuracy was determined. The evaluation of the prognosis predictive models showed that the surgery prognosis could be predicted with a high level of accuracy. The developed models can accurately predict the cases before and after surgery using a non-linear combination with complex actions, with the potential to estimate such poor prognosis as a reoperation. The models are anticipated to prove valuable to both the patients and medical staff.

Nevertheless, applying ML algorithms to a predictive analysis has certain drawbacks. Although the ML models learn the human process of judgment and make similar decisions, the basis of these results remains unknown, which is referred to as a black box. For example, when recognizing an object as a car, we are unable to explain the process by which we arrive at this conclusion.

In other words, a good prognosis may be predicted for a patient, but the doctor may not be able to find the evidence in the diagnosis to support the prediction, and it would be difficult to explain the reasoning behind the prediction to the patient. Unlike conventional statistical models, the models developed using ML algorithms are often non-linear, and thus the number of rules or parameters that define the model can reach up to several billion. Therefore, A plus B does not always produce C [16]. The precise data processing pathway of ML is a black box, whose decoding is challenging even for data scientists. The results of such a black box with the challenge of pathway decoding cannot unconditionally be trusted without validation, which is essential. These results should be validated by considering whether the given algorithm has accurately predicted a future event or result as desired and whether the outcome is useful in practice [17].

However, concepts such as local interpretable model-agnostic explanations have emerged, and they aim to overcome the previously mentioned problems [18]. The local surrogate model is used to explain the individual predicted values of the black box ML model with an additional advantage in that previous studies on interpretable models were used as local surrogate models and accumulative experience has been made available [19]. Based on such findings, a more appropriate method to utilize the ML would be to identify and apply the direction of improvement regarding the underlying mechanism with human intuition, rather than just accepting the black box model while optimizing only the accuracy of the validation dataset.

Although there are both advantages and disadvantages associated with ML technology, the proposed study is in line with those reported by Siccoli [20], Azimi [21], and Guha [22] that claim the technological possibility of various, clinically relevant prognosis predictions for the decompression surgery of spinal stenosis and the possibility of shared decision-making between the patient and the surgeon based on the predicted information. The approach to a prognosis predictive analysis prior to surgery based on the individual patient characteristics is anticipated to push forward the advancement of surgical treatment for patients with spinal stenosis.

The limitations of the proposed study are as follows: First, a subjective pain index was used such that an objective analysis could not be conducted with respect to pain, and the various parameters in the patient-reported outcome measures could not be used. In addition, an adequate amount of data could not be obtained, with the possibility of a slight reduction in the accuracy of the data. For further studies on maximizing the benefits of deep learning, a larger and more precise dataset is required, although it is considerably difficult to obtain the data from a single institution. To overcome such limitations, a multi-institutional study or a national-level consortium study should be conducted.

## Figures and Tables

**Figure 1 brainsci-10-00764-f001:**
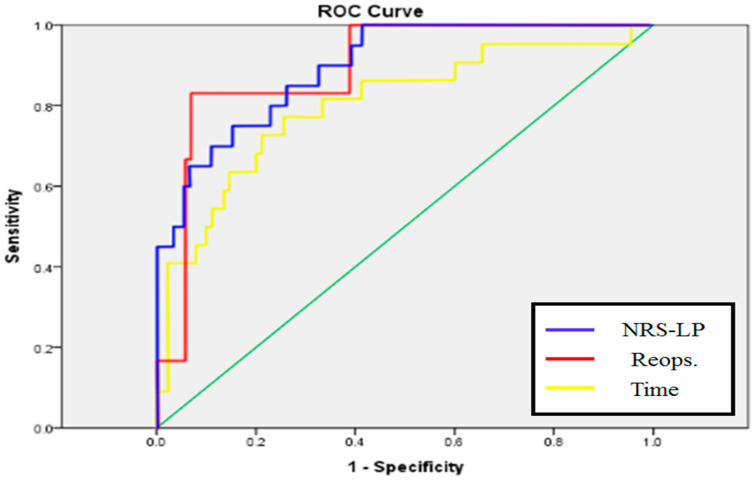
Area under the curve (AUC) result of machine-learning models for prediction of Numeric Rating Scale (NRS)-LP, reoperation, and time.

**Table 1 brainsci-10-00764-t001:** General characteristics of patients and parameters (*n* = 112)**.**

Parameters	Value
Males, n (%)	50 (44.6)
Mean age ± SD (years)	60.4 ± 12.8
Active smoker, n (%)	21 (19)
Regular alcohol intake, n (%)	56 (0.5)
The American Society of Anaesthesiologists (ASA) score, n (%)	
I	15 (13.3)
II	83 (73.5)
III	14 (13.2)
Mean height ± SD, cm	160.97 ± 8.94
Mean weight ± SD, kg	65.01 ± 12.31
Mean BMI ± SD, kg/m^2^	26.91 ± 9.74
Index level, n (%)	
L1–2	1 (0.9)
L2–3	6 (5.3)
L3–4	32 (28.5)
L4–5	72 (64.3)
L5–S1	34 (30.3)
Multi-level decompression, n (%)	32 (28.5)
Mean baseline patient-reported outcome measures (PROMs) ± SD	
Numeric Rating Scale for leg pain (NRS-LP)	7.18 ± 1.31
Medical cost affordability	
High	28 (0.25)
Moderate	72 (0.65)
Low	12 (0.10)

**Table 2 brainsci-10-00764-t002:** Summary of parameter results.

Endpoint	Number of Incident-Free Cases and Percentage	Number of Incidents and Percentage
NRS–Discharge	90 (80.4)	22 (19.6)
Reoperation		
Incidence	106	6 (5.3)
Period Parameter		
Prolonged op, >200 min	92	20 (17.9)

**Table 3 brainsci-10-00764-t003:** Estimated results of the training and evaluation datasets.

Metric	NRS Score	Reoperation	Surgery Time
Training (Bootstrapping)			
Sensitivity	84.675	94.643	89.691
Specificity	75.000	0	66.667
Accuracy	83.929	94.643	86.607
AUC	0.803	0.887	0.896
Prevalence	92.857	1	86.607
Positive predictive value	97.778	1	94.565
Negative predictive value	27.273	0	50.000
Relative risk	1.344	1	1.891
F1 score	0.9072	0.9724	0.9206
Testing			
Sensitivity	88.462	93.548	96.154
Specificity	60.000	0	90.000
Accuracy	83.871	93.548	95.161
AUC	0.860	0.952	0.975
Prevalence	83.871	1	83.871
Positive predictive value	92.000	1	98.039
Negative predictive value	50.000	0	81.818
Relative risk	1.840	1	5.392
F1 score	0.893	0.903	0.971
Brier score	0.13	0.11	0.13

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
