# Peer review of "Development of a Machine-Learning Model of Short-Term Prognostic Prediction for Spinal Stenosis Surgery in Korean Patients"

_brainsci, 2020, doi:10.3390/brainsci10110764_

Round 1
Reviewer 1 Report
Comments and Suggestions for Authors
In this manuscript, the authors attempt to develop a statistical model for the prediction of outcome after surgery for spinal stenosis. I find this work to be of potentially high interest to scientific readers. However, this manuscript will certainly benefit from a thorough review focusing on the following aspects:
Major concerns:
1. - Introduction: You refer to the observation of an increase from 1.2 to 1.6 million patients "with spinal stenosis" within 4 years. Where in the world have these data been collected? How was the diagnosis established? How were these patients treated? Please clarify.
2.3 - Empirical analysis: Which ML algorithm had the greatest explanatory power for which parameter? Please clarify.
2.3 - Empirical analysis: How was the evaluation dataset defined? Please clarify.
3. - Results: You claim that PROM, drinking, and smoking were "affected" by your prediction models. I do not understand at all how a predictor from the history of the patient, linke drinking or smoking, can be affected by a statistical model that aims to predict the outcome after surgery for spinal stenosis. In case you meant that PROM, drinking, and smoking were highly predictive of the outcome after surgery for spinal stenosis, please prove this claim (a heatmap might be helpful to have the impact of predictors on outcomes visualized at a glance).
4. - Discussion: You claim that "the ML models used in this study gave results similar to those by humans". Can you prove this?
Minor concerns:
4. - Discussion:
I find it obsolete to explain to the scientifc reader what is meant referring to a "black box".
I recommend to have this manuscript proof-read by an English native speaker.
Author Response
Dear Reviewer 1.
Thank you for your kind email. We answer your comments.
- - Introduction: You refer to the observation of an increase from 1.2 to 1.6 million patients "with spinal stenosis" within 4 years. Where in the world have these data been collected? How was the diagnosis established? How were these patients treated? Please clarify.
- The data comes from the National Health Insurance Corporation in Korea and is already famous in Korea. We have already mentioned this part in the reference and if you find it difficult to search, I attach an Internet news article.
2.3 - Empirical analysis: Which ML algorithm had the greatest explanatory power for which parameter? Please clarify.
- The most descriptive of our seven models was logistic regression and we specified that part in the manuscript and in Table 1.
2.3 - Empirical analysis: How was the evaluation dataset defined? Please clarify.
- We mentioned that 70% of the total 112 datasets were set as training datasets and 30% as evaluation datasets.
- - Results: You claim that PROM, drinking, and smoking were "affected" by your prediction models. I do not understand at all how a predictor from the history of the patient, linke drinking or smoking, can be affected by a statistical model that aims to predict the outcome after surgery for spinal stenosis. In case you meant that PROM, drinking, and smoking were highly predictive of the outcome after surgery for spinal stenosis, please prove this claim (a heatmap might be helpful to have the impact of predictors on outcomes visualized at a glance).
- We are sorry to disturb the reviewer. We deleted this part because there was an error in the editing of the article.
- - Discussion: You claim that "the ML models used in this study gave results similar to those by humans". Can you prove this?
- We revised the wording as follows.
Minor concerns:
- - Discussion:
I find it obsolete to explain to the scientifc reader what is meant referring to a "black box".
- Thank you for the reviewer's kind comment. However, we think it is desirable to explain the disadvantages of machine learning in the manuscript.
I recommend to have this manuscript proof-read by an English native speaker.
- Thank you for the reviewer's kind comment.

Reviewer 2 Report
Comments and Suggestions for Authors
The authors present a manuscript in which they attempt to verify that the individual characteristics of the patient and treatment characteristics during surgery enable a prediction of the patient's prognosis and validate the accuracy of the approach.
The authors were moderately successful in this goal. The authors utilized multiple machine learning models and picked whichever model gave them the greatest accuracy; which is not uncommon in the machine learning model literature as it relates to clinical data. However, such a methodology is not statistically sound. Typically an investigator should select whichever machine learning model is most representative of the relationship between the covariates and the outcomes that they are hoping to predict.
It is also not clear to this reviewer what the NRS was measuring or referencing. Was it referencing the patient's back pain? The patient's subjective weakness/symptoms of neurogenic claudication? Or the patient's radicular leg pain? All of which are signs and symptoms of lumbar stenosis and typically are measured independently with 1 another in patient-reported outcomes. The authors should specify what the NRS is referencing.
Author Response
Dear Reviewer 2.
Thank you for your kind email. We answer your comments.
The authors were moderately successful in this goal. The authors utilized multiple machine learning models and picked whichever model gave them the greatest accuracy; which is not uncommon in the machine learning model literature as it relates to clinical data. However, such a methodology is not statistically sound. Typically an investigator should select whichever machine learning model is most representative of the relationship between the covariates and the outcomes that they are hoping to predict.
- We totally agree with you. However, in this paper, we first performed seven representative models and selected the most descriptive model among them. Although various covariates were not selected due to the limitations of the data, this will be overcome in subsequent studies.
It is also not clear to this reviewer what the NRS was measuring or referencing. Was it referencing the patient's back pain? The patient's subjective weakness/symptoms of neurogenic claudication? Or the patient's radicular leg pain? All of which are signs and symptoms of lumbar stenosis and typically are measured independently with 1 another in patient-reported outcomes. The authors should specify what the NRS is referencing.
- The NRS used in this paper is an indicator of the radiating pain by the patient. We added this content.

Round 2
Reviewer 1 Report
Comments and Suggestions for Authors
My concerns have been addressed in a satisfactory manner.
Author Response
We appreciate the reviewer's kind understanding and review.
Reviewer 2 Report
Comments and Suggestions for Authors
Thank you to the authors for their submission
Author Response

(The authors gave the same response as above.)
